# Platelet-rich plasma for immature post-traumatic scars and early keloids: A scoping review

**Virgilio Blandón**[1]*, **Alessandro Alvarado**[2], **Miguel Borge**[3], **Erick Correa**[3], **Taylys Leyton**[2], **Sofia Gonzalez**[2]

**1** Department of Dermatology, Universidad Americana (UAM), Managua, Nicaragua, **2** School of Medicine, Universidad Americana (UAM), Managua, Nicaragua, **3** Department of Dermatology, Universidad de Ciencias Médicas y Enfermería "Dr. Julio Briceño Dávila", Managua, Nicaragua

☯ These authors contributed equally to this work.
* virgilio.blandon@uamv.edu.ni

## Abstract

### Background

Post-traumatic cutaneous scars enter an immature phase after re-epithelialization, characterized by ongoing inflammation and active remodeling. This phase, typically within six months of injury, may allow preventive interventions before pathological scarring develops. Platelet-rich plasma (PRP) has been suggested as a modulator of scar remodeling due to its growth factor content, but its role in immature scars has not been systematically reviewed.

### Objective

To map and summarize the clinical evidence on autologous platelet-containing plasma (ACP), including platelet-rich plasma (PRP), for immature post-traumatic scars and early keloids, and to identify gaps for future research.

### Methods

A scoping review was performed following established frameworks and reporting guidelines. Databases including PubMed/MEDLINE, DOAJ, SciELO, LILACS, and Google Scholar were searched. Studies evaluating ACP or PRP applied to post-traumatic scars within six months post-epithelialization were included.

### Results

Five studies involving 65 patients met inclusion criteria, including randomized trials, observational studies, and case reports. PRP was mainly delivered via intralesional injection. Evidence suggests PRP may improve clinical scar scores and patient-reported outcomes, with no serious adverse events reported. However, studies were heterogeneous in PRP preparation, dosing, timing, and outcome measures. No study was designed to assess PRP as a preventive monotherapy.

**Data availability statement:** All relevant data are within the paper and its Supporting information files.

**Funding:** The author(s) received no specific funding for this work.

**Competing interests:** The authors have declared that no competing interests exist.

## Conclusions

Evidence for PRP in immature post-traumatic scars and early keloids is extremely limited and heterogeneous. While preliminary findings indicate potential benefit and safety, current data are insufficient to support routine preventive use. High-quality, prospective studies are needed to determine whether PRP can influence scar remodeling or prevent pathological scarring.

## Introduction

The skin, as the body's primary barrier, has evolved sophisticated mechanisms to repair itself after injury. Wound healing is a complex, dynamic process that unfolds through the classical, overlapping phases of haemostasis, inflammation, proliferation, and remodelling, with the goal of restoring skin integrity and function [1]. However, in hundreds of millions of patients globally, this process results in abnormal scarring, particularly following traumatic injury, leading to substantial functional, aesthetic, and quality-of-life burdens [2].

Upon complete re-epithelialization, an immature scar forms, initiating a prolonged phase of biological activity. Cutaneous wound healing progresses through sequential, overlapping phases of inflammation, fibroplasia, and remodeling [3]. This early scar remains biologically active, representing a dynamic and malleable state [2]. High-resolution spatiotemporal analysis confirms that the cellular drivers of this plasticity, including persistent populations of migratory keratinocytes and proliferating fibroblasts, remain active well into the remodeling phase [4]. This stands in stark contrast to a mature scar, which is biologically quiescent [2,3,5], a state defined by significantly reduced cellularity, resolved inflammation, and a stabilized extracellular matrix with restored collagen cross-linking, rendering it functionally inert and less therapeutically responsive [5].

The critical period of biological immaturity and active remodeling extends for six months to a year post-injury [3]. According to international consensus, hypertrophic scars emerge from this immature state, typically becoming progressively obvious within the first 12 weeks and presenting with a growth phase of five to six months [2]. This pathophysiology underscores a critical therapeutic window within the first six months post-epithelialization. During this window, the scar remains immature and has not yet solidified into an established hypertrophic phenotype [2]. The balance between collagen synthesis and degradation is most amenable to intervention at this time, a concept experimentally supported by strategies that modulate matrix metalloproteinase (MMP) activity to reduce scarring and improve collagen architecture [6]. This plastic, pro-fibrotic microenvironment is driven by ongoing inflammation, characterized by the activity of key immune cells such as macrophages and mast cells [7]. It is therefore counterproductive that the predominant clinical strategy for immature scars often adopts a "watch-and-wait" approach, delaying targeted anti-inflammatory therapy until the scar is fully formed and the pathology is established [8]. Consequently, this immature phase represents the ideal time for a proactive, preventive strategy that targets the inflammatory drivers of fibrosis, rather than the correction of an established pathology.

Autologous platelet-containing plasma (ACP), particularly platelet-rich plasma (PRP), an autologous concentrate of platelets, has been proposed as a biologically rational candidate for such interventions. This rationale is rooted in the well-established principle that platelets serve as a rich repository of growth factors pivotal to early wound healing, including platelet-derived growth factor (PDGF) and transforming growth factor-beta (TGF-β) [1,9]. These key regulators are involved in processes such as fibroblast proliferation, collagen synthesis, and granulation tissue formation [1]. By delivering a supraphysiological concentration of these signaling molecules to the biologically active, remodeling environment of an immature scar, PRP is hypothesized to influence scar remodeling [1,10,11]. Preclinical evidence indicates that PRP can modulate persistent inflammation and may affect fibroblast activity, suggesting a possible impact on scar formation [12]. This is demonstrated by its capacity to reduce certain pro-fibrotic and pro-inflammatory mediators, such as TGF-β1, MMP-9, and NF-κB, while potentially promoting more organized collagen deposition [10,12,13]. Histological studies in humans suggest that PRP injection can induce regenerative responses in the dermis, including neocollagenesis, angiogenesis, and the formation of new adipocytes [14], although the clinical significance of these findings remains to be determined.

Through these mechanisms, PRP is hypothesized to influence scar architecture, potentially affecting the balance between hypertrophic and atrophic outcomes [15], but these effects remain speculative and clinically unproven. This proposed mechanism specifically targets the dynamic biology of the immature scar, contrasting with the static extracellular matrix of a mature scar.

Despite this compelling biological rationale and the established use of PRP in other medical fields, a critical distinction in its dermatologic application remains underexplored [10,13]. The literature has predominantly investigated PRP for its ability to accelerate the closure of open wounds [16,17], a context defined by the presence of a tissue defect [18]. Furthermore, existing reviews on PRP for scars often conflate mature and immature scars, failing to distinguish between interventions aimed at biologically active versus quiescent tissue [15].

This oversight is particularly relevant for post-traumatic scarring. The etiological origin and resulting healing microenvironment in these scars are fundamentally different from those of other types, such as acne-induced pathological scars and striae distensae. Post-traumatic scars, especially after burns, involve deep dermal injury that triggers a distinct inflammatory cascade and altered immune cell recruitment [19]. In contrast, acne scarring originates from chronic inflammation centered on the pilosebaceous unit [20]. and striae distensae result primarily from mechanical stretching and hormonal changes, leading to disruption of dermal collagen and elastin without classical inflammatory or traumatic injury [21]. These fundamental differences in pathophysiology underscore the unique biological dynamics and therapeutic considerations relevant to post-traumatic scars.

For the purpose of this scoping review, we focus on scars resulting from traumatic injury within the first six months post-epithelialization, a period that captures the immature state when the scar is biologically plastic and most hypertrophic scars have not yet fully established their phenotype (2). This temporal criterion, while designed to capture the preventive window for post-traumatic scars, also encompasses early-stage pathological scars, such as nascent keloids, that fall within the same biological timeframe of immaturity. Although keloids are traditionally classified as established fibroproliferative disorders, emerging evidence indicates that early keloids share key cellular and molecular features with immature scars, including persistently activated fibroblasts, ongoing inflammation, and pro-fibrotic signaling [22]. Therefore, these early keloids represent a biologically plausible target for preventive interventions, with potential to influence scar evolution [23], but clinical evidence remains unproven.

Thus, while our primary interest lies in the preventive potential for post-traumatic scars, we recognize that an inclusive mapping of the evidence must consider all scar types meeting this operational definition of immaturity. In stark contrast, the potential of PRP as a preventive and regenerative therapy applied specifically during this dynamic immature phase of post-traumatic scars has not been systematically examined and remains speculative. This potential specifically involves averting progression into a hypertrophic phenotype, modulating early keloid development, or enhancing the regeneration

of atrophic scars [14,15,23]. The distinction between treating the biologically active immature scar versus the quiescent mature scar represents a significant conceptual and translational gap.

Therefore, a scoping review methodology is ideally suited to systematically interrogate this emerging and likely heterogeneous body of literature [24,25]. The primary aim is to determine the volume and nature of the existing research, including its focus on both hyperproliferative and atrophic scar outcomes, and to identify critical gaps that warrant future investigation.

## Methods

### Study design and framework

This scoping review was conducted according to the methodological frameworks of Arksey & O'Malley and Levac et al. [24,25], and adhered to PRISMA-ScR guidelines [26]; the full PRISMA-ScR checklist is provided in S1 Text. The study was exploratory and mapping-focused, aiming to systematically identify and chart existing literature without testing hypotheses or involving experimental interventions, randomization, or group allocation.

### Objectives and research question

The objective of this review was to map and describe the literature on the use of ACP, including PRP, for managing recent post-traumatic scars and early keloids (≤6 months post-epithelialization), characterizing study designs, interventions, and outcomes, and identifying gaps to inform future research.

The central research question was

What is the volume and nature of evidence regarding the use of ACP, including PRP, for the treatment of immature post-traumatic scars and early keloids, and what are the reported clinical, histological, and patient-reported outcomes?

### Eligibility criteria

Studies were considered eligible if they reported on human patients with recent post-traumatic cutaneous (skin) scars and early keloids within 6 months post-epithelialization and evaluated interventions using any APC or PRP formulation administered intralesionally or topically. For this review, the operational definition of an immature scar was used, based on the international scar classification consensus, defined as an early and transient phase characterized by mild elevation, erythema, pruritus, and increased tissue density, which may normalize over time and precedes the stable maturation phase [2]. This biologically active state represents a critical window in which the eventual scar phenotype, including hypertrophic, atrophic, or keloid outcomes, remains modifiable [2]. This period of plasticity was operationally defined as occurring within 6 months following wound epithelialization, aligning with the proposed window for therapeutic intervention. Consequently, studies evaluating scars with these potential outcomes were eligible provided that the intervention was applied within this defined immature phase, with scar age and biological state serving as the primary inclusion criteria.

The target population included recent post-traumatic cutaneous scars, encompassing both accidental trauma and surgical incisions, while excluding acne scars and striae distensae, which have a distinct pathophysiology and remodeling process [20,21]. This ensured inclusion of scars with a clearly defined onset suitable for preventive interventions during the biologically active immature phase.

While recognizing the clinical distinction between hypertrophic scars and keloids, early-stage keloids (nascent or forming) were included because they share key biological features with immature scars, including active inflammation and a theoretically modifiable remodeling process, and thus represent a plausible target for preventive interventions [22,23]. This ensures methodological consistency and aligns with the review's focus on the dynamic, plastic phase of scar development.

Eligible study designs included peer-reviewed primary research such as case reports, case series, observational studies, and clinical trials. Studies were required to report clinical, histological, or patient-reported outcomes related to scar appearance, symptoms, or tissue remodeling.

 

Studies were excluded if they focused exclusively on mature scars beyond 6 months post-epithelialization, involved non-human subjects, evaluated wounds rather than established scars, relied on preclinical or in vitro models, were conference abstracts or preprints, or were not peer-reviewed. Studies limited to pediatric populations were also excluded, as fundamental biological differences in wound healing and scar remodeling exist between children and adults. Experimental and developmental evidence indicates that fetal and pediatric skin exhibits a more regenerative phenotype, characterized by accelerated re-epithelialization and distinct cellular and molecular responses compared with adult skin [27,28]. Inclusion of pediatric studies would therefore introduce substantial biological heterogeneity and potentially confound the interpretation of findings specific to adult scar pathophysiology, which was the focus of this review. Studies lacking sufficient data for extraction were excluded. No language restrictions were applied, allowing for comprehensive data extraction and broad international representation.

### Search strategy

A comprehensive electronic search was conducted to identify studies evaluating PRP and related autologous platelet-derived products for immature post-traumatic scars and keloids. Searches were performed in PubMed/ MEDLINE, DOAJ, SciELO, LILACS, and Google Scholar. The search strategies combined controlled vocabulary (e.g., MeSH terms) and free-text terms covering platelet concentrates, PRP, scar types, tissue repair, and temporal parameters including early, recent, acute, and immature scars. Boolean operators were applied to maximize sensitivity.

All records retrieved from the databases were imported into Rayyan, for initial deduplication and preliminary screening. Following this automated step, a manual review was performed to ensure complete removal of duplicates and verify dataset accuracy prior to full-text screening. This two-step process ensured a rigorous and reliable selection of studies. The full search strategies and results for each database are presented in S1 Table.

### Study selection

After deduplication, which was performed by a single reviewer and subsequently verified by a second reviewer, all records underwent title and abstract screening against the predefined eligibility criteria. The full texts of potentially relevant studies were then assessed for inclusion. Both the title/abstract and full-text screening phases were conducted independently by two reviewers. Disagreements were resolved through consultation with a third reviewer to reach consensus. The results of the deduplication, screening, and article inclusion/exclusion process are summarized in Supporting information S2 and S3 Tables, which provide counts of duplicates removed, unique records screened, exclusions at each stage, and final articles included for full-text review. This process ensured a rigorous and unbiased selection of studies for inclusion in the scoping review.

### Citation tracking

Citation tracking was conducted iteratively to ensure comprehensive coverage. Backward and forward citation searching was performed for all studies ultimately included in the review to identify additional relevant publications. For review articles that were excluded solely because they were reviews, backward citation searching was performed to capture potentially eligible primary studies. Whenever a new eligible study or review was identified, backward and forward citation searching was also conducted for that source. This process continued exhaustively until saturation was reached, defined as the point at which all included studies, all relevant reviews, and all publications cited within these sources that met the inclusion criteria had been examined and no additional eligible studies could be found. A detailed summary of the backward and forward citation searches, including source articles, total results retrieved, articles reviewed, and reasons for exclusion, is provided in S4 Table.

## Data charting and evidence synthesis

Data from all included studies were charted using a predefined extraction template capturing study design, population characteristics, scar type and age, ACP/PRP modality and route of administration, number of sessions, outcome measures, and key findings. The initial data extraction was performed by one reviewer and subsequently verified by a second reviewer to ensure accuracy and consistency. Variables, definitions, and instructions for extraction are provided in S5 Table, which served as a standardized data dictionary to guide reviewers. Synthesis was descriptive and mapping-focused, summarizing study characteristics, interventions, and outcomes in both narrative and tabular forms. Gaps in the literature were highlighted to inform future research priorities.

## Risk of bias assessment

A risk of bias assessment was performed for all included studies. The purpose was descriptive and aligned with the scoping review objective to map and characterize the available evidence; it was not used to exclude studies or to draw inferential conclusions about efficacy [24–26]. Design-specific tools were applied: the Cochrane RoB 2 tool for randomized trials [29], the ROBINS-I (v2) tool for the non-randomized interventional study [30,31], and the relevant Joanna Briggs Institute (JBI) critical appraisal checklists for case reports and series [32,33].

One reviewer conducted the initial assessment using these tools, which was then verified in full by a second reviewer. Any disagreements in judgments were resolved through discussion and consensus between the two reviewers. The outcomes of this process were synthesized narratively to describe recurring methodological patterns and limitations across the evidence base, such as sample size constraints or variability in intervention protocols. This approach is consistent with methodological guidance for scoping reviews, where critical appraisal is an optional component used to contextualize the mapped literature.

## Protocol registration

The protocol for this scoping review was preregistered on the Open Science Framework (OSF) under the title *'Mapping the Evidence on Platelet-Rich Plasma for Recent Post-Traumatic Scars: A Scoping Review Protocol'* and is available at https://osf.io/y6wg2/overview (https://doi.org/10.17605/OSF.IO/Y6WG2). The preregistration details the study objectives, eligibility criteria, search strategy, study selection, data charting, and evidence synthesis [34]. Any deviations from the preregistered protocol are reported transparently in the review.

## Timeline

The initial electronic literature search was completed on November 11th, 2025. The first round of screening was completed on December 26th, 2025. Following this, citation tracking and review of newly identified studies were conducted through December 30th, 2025. Data extraction and charting were completed on 10 January 2026, and the writing of the results section was finalized on 13 January 2026. This timeline reflects the sequential completion of all methodological steps and ensures transparent reporting of the review process.

## Results

### Search results and study selection

The comprehensive search of five electronic databases, supplemented by backward and forward citation tracking, identified a total of 1,325 records. After removal of 79 duplicates, 1,246 unique records were screened at the title and abstract level according to the predefined eligibility criteria. Of these, 1,213 records were excluded, most commonly because the study population involved scars beyond the predefined threshold of six months after epithelialization (n = 972). 33 articles were subsequently retrieved and assessed in full text for eligibility.

During full-text assessment, 28 articles were excluded, all primarily due to failure to meet the population criterion, as the scars evaluated exceeded the six-month post-epithelialization window. One disagreement between reviewers regarding eligibility was resolved through consultation with a third reviewer [35]. Ultimately, five studies met all inclusion criteria and were included in the final qualitative synthesis [35–38]. The complete study selection process is illustrated in the PRISMA-ScR flow diagram (Fig 1). This highlights a striking paucity of clinical research evaluating ACP specifically for immature post-traumatic scars, including keloids. Despite screening over 1,300 records, only five studies met inclusion criteria, underscoring a significant evidence gap in early-stage scar management.

This figure illustrates the identification, screening, eligibility assessment, and inclusion of studies evaluating ACP, including PRP, for immature post-traumatic scars and early keloids. A total of 1,325 records were identified through database searching and citation tracking. After removal of duplicates, 1,246 records were screened, 33 full-text articles were assessed for eligibility, and five studies met the inclusion criteria and were included in the qualitative synthesis.

## Characteristics of included studies

Five studies met the inclusion criteria for this scoping review, representing a spectrum of study designs: one single-blind randomized comparative clinical trial [37], one randomized controlled trial [35], one prospective observational cohort study

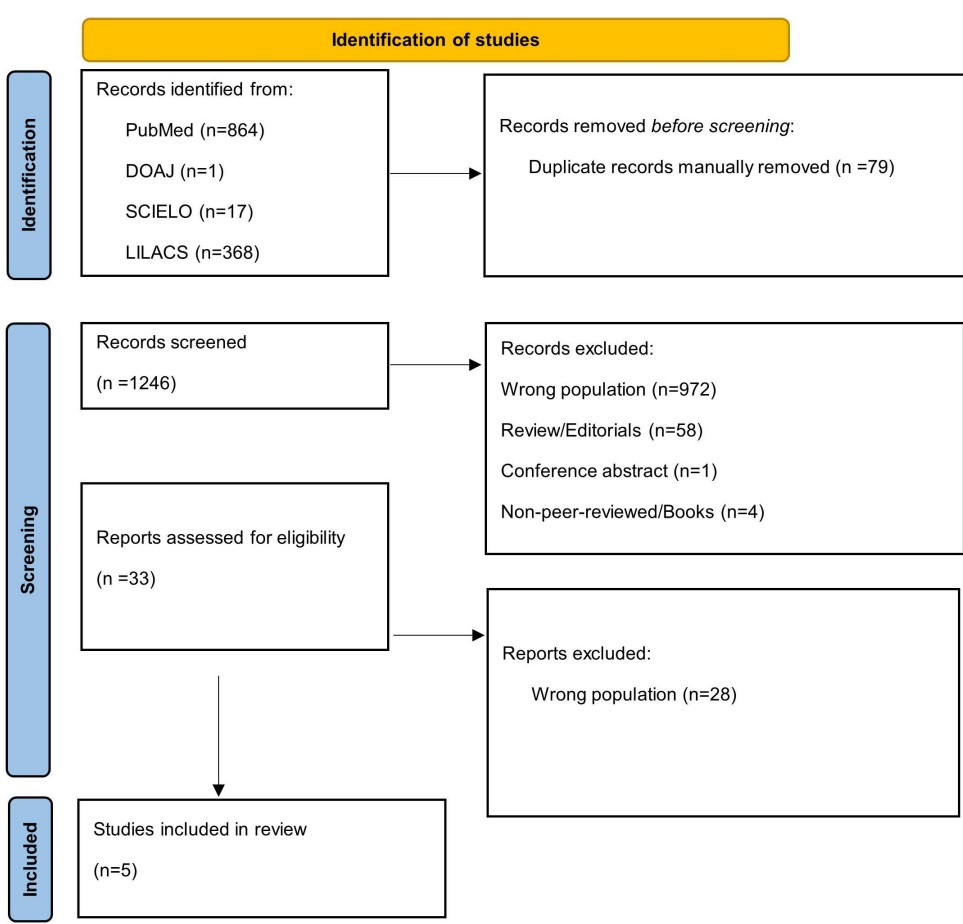

**Fig 1. PRISMA-ScR flow diagram of study selection.**

with mixed interventions [39], one case series [38], and one single-patient case report [36]. The included studies were published between 2018 and 2024.

Across all included studies, sample sizes for patients treated with autologous PRP ranged from one to 40 participants, resulting in a cumulative total of 65 patients receiving PRP. The target populations primarily consisted of patients with early or immature scars, including keloids, post-burn scars, and post-traumatic facial scars, generally within six months of injury or epithelialization.

Intralesional injection was the predominant route of PRP administration in all studies. PRP was evaluated as monotherapy in two randomized trials [35,37], while the remaining studies investigated PRP exclusively as part of multimodal regenerative or non-surgical treatment strategies [36,38,39]. Concomitant therapies varied and included intralesional corticosteroids, botulinum toxin type A, polynucleotide-based injectables, fractional $CO_2$ laser therapy, hyaluronic acid fillers, silicone gel, and pressure therapy.

Marked heterogeneity was observed in PRP preparation and administration protocols, including differences in centrifugation methods, platelet concentration, activation techniques, injection volumes, treatment intervals, and number of sessions. Critically, an evaluation of the available data reveals that only one study provided sufficient detail on PRP composition to allow for a minimal functional categorization, specifying the use of leukocyte-rich PRP with calcium gluconate activation [36]. The remaining four studies did not specify leukocyte content or platelet concentration, and details regarding PRP activation were either omitted or unclear [35,37–39]. Consequently, a consistent categorization of PRP formulations across the included studies was not feasible. Outcome assessment methods were similarly diverse. Most studies employed validated clinical scar assessment tools, most commonly the Vancouver Scar Scale (VSS) and/or the Patient and Observer Scar Assessment Scale (POSAS). Several studies supplemented these scales with additional evaluation modalities such as dermoscopy, histopathological or immunohistochemical analysis, standardized photographic documentation, and patient-reported outcomes.

Follow-up duration varied considerably across studies, ranging from several weeks to ten months. Reporting of long-term outcomes and recurrence was limited. Due to substantial clinical and methodological heterogeneity, as well as frequent use of combination therapies, direct comparison across studies was not feasible.

A brief summary of the key characteristics and principal outcomes of the included studies is presented in Table 1, while the complete detailed data including PRP preparation, dosing, route, adjunct therapies, quantitative efficacy data, and study limitations is available in S6 Table.

Definitions of scar chronicity were inconsistently reported and rarely operationalized across studies. In keloid research, only Neinaa et al. explicitly included newly diagnosed, untreated keloids with a duration of less than six months [37]. Albalat et al. did not use lesion duration as an inclusion criterion, although the mean duration of keloids among enrolled patients was approximately four months [35]. None of the keloid studies distinguished between clinically relevant temporal benchmarks, such as time since initial epithelialization, time since the inciting injury, or time since keloid formation, each representing a distinct stage in scar pathophysiology. Among post-traumatic scars, only El-Orabi et al. explicitly referred to "immature" scars, Ruiz et al. described interventions as "early" without defining temporal parameters, and Kim et al. reported time since injury for each scar but did not clarify whether the lesions were actively remodeling or largely stabilized [36,38,39]. Importantly, none of the studies considered how ongoing scar maturation might influence treatment outcomes, which complicates the distinction between effects attributable to PRP therapy and changes resulting from the natural healing process. Table 2 provides a study-level synthesis of scar type, timing, morphology, and assessment methods, underscoring the need for standardized temporal and morphological criteria in future research.

## Risk of bias appraisal

A descriptive assessment of methodological quality was performed to map the evidence and contextualize study findings [24–26]. Risk of bias was evaluated using design-specific tools: RoB 2 for randomized controlled trials [29], ROBINS-I for the non-randomized study [30,31], and JBI checklists for case series and case reports [32,33].

**Table 1. Summary of included studies evaluating autologous PRP for early or immature scars.**

| Author, Year | Study Design & Population/ PRP Subgroup | Scar Type/ Time Since Injury | PRP Intervention/ Sessions | Outcome Measures/ Follow-up | Main Findings |
|---|---|---|---|---|---|
| Ruiz et al., 2018 | Case report: 1 female patient/ PRP 1 | Post-burn hypertrophic/ 7 weeks | Topical + intralesional leukocyte-rich PRP/ 6 intralesional applications + daily topical for 5 days per session | VSS, POSAS, photography/ 10 months | Marked improvement in scar pliability, vascularity, pain, pigmentation |
| Neinaa et al., 2021 | Single-blind randomized comparative trial: 60 patients/ PRP 20 | Keloids/ < 6 months | Intralesional PRP monotherapy/ 3 sessions, 0.1 mL per injection point | VSS, VRS, dermoscopy, histology/ ~12 weeks | PRP significantly improved clinical and histologic parameters; superior to triamcinolone |
| Albalat et al., 2021 | Randomized controlled trial: 160 patients/ PRP 40 | Keloids/ ~4 months | Intralesional PRP monotherapy/ 6 sessions, 3-week intervals | POSAS, clinical evaluation/ 30 weeks | PRP achieved meaningful scar improvement with low recurrence and minimal adverse effects |
| El-Orabi et al., 2022 | Prospective case series: 15 patients/ PRP 1 | Immature facial scars/ ≤ 6 months | Intralesional PRP as adjunct to $CO_2$ laser/ 1 session | POSAS, photography/ up to 12 months | PRP efficacy could not be independently assessed due to limited exposure |
| Kim et al., 2024 | Case series: 8 patients/ PRP 3 | Post-traumatic, post-operative, burn, iatrogenic/ 1–6 months | Intralesional PRP as adjunct to polynucleotides ± botulinum toxin/ 1–10 sessions | Clinical assessment, patient-reported outcomes, photography/ 1–3 months | Multimodal therapy including PRP improved scar appearance and skin quality; PRP may have synergistic effects |

Note. Studies are presented in chronological order. Columns summarize study design and population with PRP subgroup, scar type and time since injury, PRP intervention and number of sessions, outcome measures with follow-up duration, and the main findings. PRP was administered either as monotherapy or as part of multimodal regenerative treatments. VSS: Vancouver Scar Scale; POSAS: Patient and Observer Scar Assessment Scale; VRS: Verbal Rating Scale.

Overall, the methodological quality of included studies was heterogeneous. Randomized trials demonstrated moderate internal validity, with limitations primarily related to incomplete blinding and absence of pre-specified analysis plans. The non-randomized study was at high to critical risk of bias, largely due to confounding by indication and unblinded outcome assessment. Case series and reports provided detailed descriptive data suitable for mapping populations, interventions, and outcomes, but were inherently limited in their ability to support causal inference or generalizable conclusions.

Consistently across the evidence base, small sample sizes, variable intervention protocols, and subjective outcome measures contributed to methodological limitations. No studies were excluded based on risk of bias, in keeping with the exploratory and mapping purpose of this scoping review [24–26].

A summary of risk of bias assessments is presented in Table 3. Detailed appraisals for each study are provided in S7–S11 Tables, including JBI checklists for case reports and series (S7–S8), Cochrane RoB 2 for RCTs (S9–S10), and ROBINS-I for the non-randomized study (S11).

## Synthesis of findings

The included studies provided limited and heterogeneous evidence based on the use of APC, including PRP, for immature post-traumatic scars. Interventions were applied primarily within six months of injury or epithelialization and were generally associated with improvements in scar appearance, texture, and patient-reported outcomes, with no serious adverse events consistently reported.

Randomized trials offer the highest level of evidence. Neinaa et al. (2021) reported outcomes of PRP comparable to botulinum toxin and superior to triamcinolone in early keloids, including favorable histopathological changes [37]. Albalat et al. (2022) found that PRP reduced POSAS scores similarly to triamcinolone, with superior outcomes compared with 5-fluorouracil, but inferior to verapamil [35]. These findings indicate potential clinical benefit for early-stage scars, while highlighting variability in protocols, short follow-up, and limited standardization.

 

**Table 2. Scar characteristics and assessment methods in PRP studies.**

| Study | Scar Type/ Time Since Injury | Morphology/ Clinical Features | Assessment Methods |
|---|---|---|---|
| Aura Ruiz et al., 2018 | Second-degree burn scar/ 7 weeks after VASER liposculpture (early intervention) | Right abdominal flank (10 × 8 cm) Hypopigmented; irregular; erythematous-violaceous Hypertrophy in upper pole Atrophy in middle area Contracture in lower posterior pole Fitzpatrick IV phototype | VSS initial: 11 points POSAS initial: 49 points Evaluated by internal and external observers Photographic documentation |
| Albalat et al., 2021 | Keloids/ Mean duration 4 months | Keloids mostly on head and neck (1–3 lesions per patient) Size 1–15 cm Fitzpatrick II–V phototype Both sexes Ages 10–60 | POSAS baseline and end-of-treatment measurement (24 weeks) ≥50% reduction in POSAS = effective response Side effects: erythema 35%, pain 100%; no recurrence reported |
| Neinaa et al., 2021 | newly diagnosed untreated stable keloids/ Duration <6 months | Head, neck, trunk, extremities 1–3 lesions per patient Size 1–5 × 0.5–4 cm Fitzpatrick II–V Both sexes, ages 12–56 Spontaneous, post-burn, post-traumatic Vascularity, pigmentation, pliability, height variable | VSS before and after treatment VRS for pain and itching Dermoscopic examination Histopathology (H&E) and immunohistochemistry for CTGF expression Clinical improvement categorized: excellent (>75%), moderate (50–75%), mild (25–50%), poor (1–25%), no response (0%) |
| El-Orabi et al., 2022 | Immature facial scar/ 3–4.5 months | Diffuse scar on face Variable location (cheek, mental area, neck) Hypertrophic Early post-traumatic or post-burn Patient age 29 Female | PRP as adjuvant to fractional CO2 laser (FCL) POSAS Photographic documentation Assessment at 6 months and follow-up to 12 months |
| Kim et al., 2024 – Case #4 | Second-degree chemical burns from battle injuries/ 4 months post-injury | Male patient Extensive burns on right side of body, left arm, and right ear Hypertrophic; erythematous Extensive tissue damage | PRP (1 session) Rejuran Healer (10 sessions) Incobotulinum toxin A (150 units) Photographic documentation Follow-up at 1 month post-treatment |
| Kim et al., 2024 – Case #5 | Traumatic facial scars from explosion/ 1 month post-injury | Female patient Multiple facial scars; elevated Irregular texture Discoloration Hypertrophic | PRP (10 sessions) combined with six Rejuran S sessions (6 cc) Bi-monthly Incobotulinum toxin A (100 units) Photographic documentation Clinical observation of scar reduction and skin rejuvenation |
| Kim et al., 2024 – Case #8 | Lip wound from dog bite/ 3 months post-injury | Female patient; lip scar; post-traumatic; mild hypertrophy and irregularity | One PRP session intercalated between two Rejuran HB plus sessions Photographic documentation Evaluation of wound healing and scar appearance |

Note. This table provides a study-level overview of scar type, timing since injury, morphology, clinical features, and outcome assessment methods. It highlights the heterogeneity in scar etiology, appearance, and evaluation approaches, including both objective measures (VSS, POSAS, VRS) and additional methods such as dermoscopy, histopathology, immunohistochemistry, and photographic documentation. The table also specifies the combination of PRP with adjunctive treatments, including polynucleotide-based injectables, fractional $CO_2$ laser therapy, and botulinum toxin, where applicable. CTGF: connective tissue growth factor; VASER: vibration amplification of sound energy at resonance.

Observational and descriptive studies illustrate the use of PRP in combination with other therapies. Kim et al. (2024) and Ruiz et al. (2024) documented subjective improvements in scar appearance and texture, while El-Orabi et al. (2022) demonstrated PRP applied adjunctively with laser and silicone gel, reflecting real-world multimodal approaches rather than monotherapy [36,38,39].

**Table 3. Summary of risk of bias assessments for included studies.**

| Study | Design | Tool | Key Domains/ Judgement | Overall Risk of Bias |
|-------|--------|------|------------------------|----------------------|
| Ruiz et al., 2018 | Case report | JBI Checklist | Patient demographics: Yes; History: Yes; Clinical condition: Yes; Diagnostic assessment: Yes; Intervention: Yes; Post-intervention outcome: Yes; Adverse events: No; Takeaway lessons: Yes | Moderate (descriptive, limited generalizability) |
| Kim et al., 2024 | Case series | JBI Checklist | Inclusion criteria: Yes; Condition measurement: Unclear; Valid identification: Yes; Consecutive inclusion: No; Complete inclusion: No; Demographics: Yes; Clinical info: Yes; Outcomes: Yes; Site info: Yes; Statistical analysis: No | Moderate (descriptive, non-consecutive, no stats) |
| Albalat et al., 2022 | RCT | Cochrane RoB 2 | Randomization: Some concerns; Deviations from interventions: Some concerns; Missing data: Some concerns; Outcome measurement: Some concerns; Selective reporting: Some concerns | Some concerns |
| Neinaa et al., 2021 | RCT | Cochrane RoB 2 | Randomization: Low risk; Deviations: Some concerns; Missing data: Low risk; Outcome measurement: Low risk; Selective reporting: Some concerns | Some concerns |
| El-Orabi et al., 2022 | Prospective cohort | ROBINS-I | Confounding: Critical; Classification: Moderate; Selection: Low; Missing data: Low; Outcome measurement: Serious; Selective reporting: Moderate | Critical |

Note. This table presents the overall and domain-level judgments for each study using design-specific tools: JBI Critical Appraisal Checklists for case reports and case series, Cochrane RoB 2 for randomized controlled trials, and ROBINS-I for non-randomized studies. Key methodological strengths and limitations are summarized, and the highest level of concern per study is reflected in the overall risk of bias. Detailed rationale and full domain-level assessments are provided in S7–S11 Tables. Abbreviations: JBI, Joanna Briggs Institute; RoB 2, Risk of Bias 2; ROBINS-I, Risk of Bias In Non-randomized Studies of Interventions; POSAS, Patient and Observer Scar Assessment Scale; VSS, Vancouver Scar Scale; TAC, triamcinolone; BTX-A, botulinum toxin type A; FU, follow-up; AE, adverse events; NA, not applicable.

Overall, the evidence suggests a potential therapeutic window during the immature phase of scar formation within six months post-epithelialization. However, heterogeneity in PRP preparation, dosing, timing, outcome measures, and concomitant treatments limits comparability and prevents definitive conclusions. No studies were specifically designed as primary trials of PRP monotherapy for immature scars, indicating a notable gap in early preventive interventions.

Preliminary evidence supports PRP as a potentially beneficial intervention in early-stage scar management, but the literature remains exploratory and methodologically diverse. Well-designed prospective studies with standardized protocols, clearly defined early scar populations, and consistent outcome measures are needed.

## Discussion

This scoping review was undertaken to systematically map the evidence for autologous platelet-rich plasma (PRP) as a therapeutic intervention targeting the immature, biologically active phase of post-traumatic scars and early keloids (≤6 months post-epithelialization) [2,3]. A central finding is the marked discordance between a strong biological rationale, supported by experimental and translational studies demonstrating that PRP can modulate key fibrotic pathways involved in scar formation, including connective tissue growth factor signaling [40], and by comprehensive mechanistic reviews describing the broad spectrum of regenerative growth factors present in PRP [41,42], and the near-absent clinical evidence base focused on this early stage of scar evolution. Only five eligible studies were identified among more than 1,300 screened records, providing empirical support for the hypothesized translational gap between the emerging concept of proactively modulating a dynamic, plastic scar and the prevailing clinical paradigm, which is characterized by a predominantly "watch-and-wait" approach before targeting inflammation [8], and is reflected in the literature on immature scar management, where evidence for preventive modalities is often limited or of low quality [43]. Although the available evidence is sparse and exploratory, it suggests that PRP is well tolerated and may be associated with improvements in early scar characteristics; however, these findings are preliminary and do not establish clinical efficacy, providing preliminary support for further investigation in rigorous, prospective studies.

## Principal interpretation: Confirmation of a paradigm gap

The primary objective of this scoping review was to characterize the extent and nature of the existing evidence on ACP and PRP for immature post-traumatic scars and early keloids. The findings demonstrate that this evidence base is extremely limited and predominantly exploratory, with the vast majority of excluded articles investigating scars beyond the six-month post-epithelialization period. This distribution reflects a prevailing emphasis in both clinical research and practice on established, mature scars rather than on the immature, biologically active phase of scar development.

This pattern provides empirical support for the "watch-and-wait" paradigm identified in the introduction [8], in which active intervention is typically deferred until pathological scarring is clinically evident. Even studies that technically met the temporal inclusion criteria, such as those including keloids under six months' duration [35,37], treated these lesions as already established pathological entities rather than as dynamic, potentially modifiable scars. The present findings reveal that the evidence base is not only limited but also conceptually misaligned with the question of therapeutic timing. Although PRP was applied within the first six months post-epithelialization, the inconsistent operationalization of scar chronicity, ranging from explicit duration-based inclusion to vague descriptions of early intervention, reflects a fundamental problem. The field lacks a shared framework for defining an immature scar as a target for preventive therapy. This conceptual ambiguity prevents meaningful analysis of dose-timing relationships, as the underlying variable, the stage of scar maturation, was neither uniformly defined nor systematically measured. Consequently, the available evidence cannot determine whether a therapeutic window exists, or when it might open or close. From this perspective, the gap is not merely empirical but ontological, as the category of immature scar has not been stabilized as a researchable object across studies.

Despite the lack of direct comparative data, the underlying pathophysiology supports discussion of a biologically plausible window. During the early remodeling phase, scars exhibit sustained cellular activity, active fibroblast populations, and ongoing extracellular matrix reorganization [2,3,5,44]. Experimental evidence indicates that PRP can modulate inflammatory mediators, influence profibrotic signaling pathways, and promote more organized collagen architecture [10,12,13], while histological studies suggest induction of regenerative processes such as neocollagenesis and neoadipogenesis [14]. These mechanisms are most relevant in a tissue environment that retains structural plasticity and biochemical responsiveness, conditions characteristic of early remodeling.

The temporal boundaries of such a window may be inferred from the natural history of scar maturation. Hypertrophic scars typically become clinically apparent within the first twelve weeks after injury and may enter a proliferative phase lasting five to six months [2]. This trajectory suggests that the period of greatest therapeutic opportunity may lie within the first three months following complete epithelialization, when fibrotic remodeling is still evolving and potentially reversible. As extracellular matrix architecture stabilizes, cellularity decreases, and inflammatory activity subsides [2,3,5], responsiveness to biologically targeted interventions is likely to diminish, consistent with the recognized therapeutic resistance of mature scars [5].

Taken together, these considerations delineate a biologically plausible therapeutic window for PRP that aligns with the understanding of hypertrophic scars and keloids as fibroproliferative disorders arising from dysregulated wound healing [45]. However, this construct remains hypothetical and derived from pathophysiological reasoning rather than prospective comparative data. Well designed studies that systematically vary the timing of PRP administration are required to empirically define the boundaries of this window.

Overall, this review delineates a clear limitation in the current literature and confirms that the role of PRP as a proactive intervention during the immature phase of scar development remains largely unexplored. The available evidence does not permit conclusions regarding clinical efficacy or optimal timing, underscoring the need for rigorously designed trials specifically targeting this early and potentially modifiable stage of scar maturation.

## Synthesis in context: Plausibility, promise, and heterogeneity

The included studies generally support biological plausibility for PRP as a modulatory intervention during the immature phase of scar remodeling, although the evidence remains limited in scope and methodological rigor. Among controlled studies, Neinaa et al. (2021) provide the clearest mechanistic evidence, reporting a significant reduction in connective tissue growth factor (CTGF) expression alongside favorable histological and clinical changes in early keloids treated with intralesional PRP [37]. Albalat et al. (2021) further compared intralesional PRP with triamcinolone, verapamil, and 5-fluorouracil in a clinical trial, showing that PRP achieved efficacy comparable to triamcinolone and superior to 5-fluorouracil, with minimal side effects and low recurrence rates [35]. This finding aligns with the concept of PRP as a potential antifibrotic modulator during active scar remodeling [46].

In contrast to controlled trials, case reports and small series describe preliminary associations between PRP application and improvements in early scar characteristics. Ruiz et al. (2018) observed improvements in pliability, vascularity, pain, and pigmentation in a post-burn hypertrophic scar treated within weeks of epithelialization [36]. Kim et al. (2024) reported potential improvements in scar appearance and skin quality across heterogeneous post-traumatic and post-operative scars treated within six months of injury [40]. El-Orabi et al. (2022) described favorable outcomes in an immature facial scar treated with PRP as an adjunct to fractional $CO_2$ laser therapy [39]. While consistent with biological responsiveness of immature scars, these observations cannot distinguish treatment effects from natural scar maturation and should therefore be interpreted cautiously. The absence of quantitative PRP characterization also limits attribution of outcomes to specific product properties [47,48].

Methodological heterogeneity further constrains interpretation. PRP interventions varied in preparation protocols, platelet concentration, leukocyte content, activation methods, delivery schedules, and timing relative to injury. Outcome assessment was inconsistent, relying on diverse clinical scales, follow-up durations, and selective histological or dermoscopic endpoints. Although validated tools such as POSAS and VSS are widely used, their subjective nature and limited sensitivity to early changes restrict their utility for preventive interventions [49,50]. Objective measures including scar area, volume, height, and depth were frequently omitted despite their importance for comprehensive assessment [2]. The absence of within-patient comparators is particularly relevant, as improvements may reflect spontaneous remodeling rather than a true PRP effect. Intra-patient controlled designs would strengthen causal inference [51], as demonstrated in incisional negative pressure wound therapy trials [52].

Technical variability in PRP preparation compounds this clinical heterogeneity. Formulations labeled identically may represent biologically distinct products, as differences in platelet dose, leukocyte content, growth factor profiles, and platelet recovery rates directly influence delivered dose and efficacy, with recovery ranging from below 70% to optimized protocols reaching 95.7% [47,48,53]. Most included studies did not report quantitative compositional data or details regarding PRP activation protocols, limiting reproducibility and mechanistic interpretation [35,37,39,40].

Experimental evidence provides a biological explanation for why such variability matters. Total platelet dose and enrichment ratio modulate growth factor delivery, fibroblast activity, and angiogenesis [48,53]. A higher total platelet dose has been associated with improved outcomes in knee osteoarthritis [48], whereas in rodent skin wounds a moderate enrichment ratio outperformed a high ratio in promoting fibroblast proliferation, angiogenesis, wound closure, and inflammation reduction [53]. Proteomic analyses further demonstrate that technical differences translate into distinct cellular responses. Platelet-rich fibrin contains higher leukocyte concentrations than first-generation PRP yet induces a comparable anti-inflammatory M2-like macrophage polarization while enhancing pro-angiogenic protein expression [54]. Leukocyte-depleted plasma preferentially activates proliferative and growth factor pathways, whereas leukocyte-rich preparations induce inflammatory signaling programs [55]. Even products with broadly similar proteomic profiles can diverge biologically, as autologous serum selectively upregulated inflammatory and pro-fibrotic pathways including mTOR and TGF-β1

signaling in corneal keratocytes [56]. Together, these findings indicate that subtle compositional differences may shift fibrotic signaling trajectories, complicating cross-study comparison.

The evidence base remains constrained by the small number of controlled trials, limited follow-up, and moderate risk of bias. No included study was specifically designed to test PRP as a preventive monotherapy for immature scars. Most applications occurred within multimodal strategies or in scars already exhibiting pathological features, reflecting heterogeneous clinical implementation without mechanistic alignment. Notably, only two randomized trials evaluated PRP as monotherapy [35,37], whereas the remaining studies incorporated it into combination regimens alongside laser therapy, injectables, or other adjunctive modalities [36,38,39]. While this approach mirrors routine clinical practice, it substantially limits causal inference, as observed improvements may reflect the concomitant intervention, synergistic effects, or natural scar maturation rather than the independent contribution of PRP [15]. In the absence of controlled comparisons that isolate PRP while holding other treatments constant, its specific therapeutic value remains uncertain.

Overall, the available evidence supports biological plausibility and suggests exploratory clinical potential but remains insufficient to define efficacy or guide practice. Any apparent benefit should therefore be considered hypothesis-generating rather than confirmatory. Advancing the field will require standardized reporting of PRP preparation and composition, clearly defined immature scar populations, biologically aligned trial designs, and sensitive longitudinal outcome measures that account for the prolonged course of scar maturation [2,44,47]. Objective morphometric metrics and tissue biomarkers should complement clinical scales. Optimization of dosing strategies will likely depend on both total platelet dose and local enrichment ratio, which appear to be tissue-specific determinants of response [48,53].

## Implications for research and clinical practice

Based on the gaps identified in this review, four methodological priorities emerge for future research:

First, rigorous characterization of scar type and strict adherence to a consensus based definition of immaturity are essential prerequisites for meaningful clinical investigation. Although the international scar management consensus [57] and subsequent literature [2] recognize the immature scar as a critical window for preventive intervention, the field has yet to operationalize this concept in a reproducible manner. Future studies must therefore adopt an explicit and standardized definition of scar immaturity, ideally anchored to established classification frameworks and the biologically plastic phase within six months of complete epithelialization [2,44]. Based on this definition, investigators should clearly report the interval between injury, complete epithelialization, and initiation of treatment. Without precise temporal documentation, it is impossible to determine whether an intervention is applied during the modifiable phase or after maturation has begun, thereby obscuring whether the strategy is truly preventive or merely therapeutic. Within this framework, prospective randomized controlled trials should enroll patients with recently epithelialized post traumatic scars and compare a standardized intralesional PRP protocol with placebo or established standard care such as silicone gel.

Second, PRP preparation and composition must be transparently and quantitatively reported to allow meaningful interpretation of outcomes. Valid comparison across studies requires documentation of three core variables: absolute platelet count, method of platelet activation, and leukocyte content. To ensure methodological clarity, investigators should report absolute platelet concentration in platelets per microliter rather than fold increase alone, total and differential leukocyte counts when relevant, activation method whether exogenous or endogenous, and complete centrifugation parameters including speed, duration, and number of spins. Adoption of structured reporting frameworks, such as the Platelet Activation White cell classification system (PAW) [58], provides a practical approach to standardizing PRP characterization, thereby enhancing reproducibility and enabling reliable cross study comparisons and meta analyses [47,48,53,59].

Third, outcome assessment must integrate validated clinical scales with objective quantitative methodologies to comprehensively characterize scar morphology. Instruments such as POSAS and the VSS provide structured clinical evaluation of scar features and remain essential components of clinical assessment [49,50]. Given that regenerative interventions such as PRP may influence dermal architecture, volumetric restoration, and biomechanical properties [14,15],

future studies should incorporate complementary instrumental and digital modalities to more precisely characterize these multidimensional changes. These may include high frequency ultrasound and elastography to assess dermal thickness and subcutaneous architecture, three-dimensional imaging systems to quantify contour, surface area, and volume, and standardized digital analysis of calibrated photographs to evaluate morphologic and colorimetric parameters [60–64]. Assessments should be performed in a blinded manner at predefined intervals, particularly at 6 and 12 months, to adequately document the critical remodeling phase and the full temporal trajectory of scar maturation [2,44].

Fourth, study design should favor intra individual randomized controlled approaches whenever anatomically feasible. Scar evolution is influenced by numerous patient, wound, and procedural factors, and immature scars may undergo spontaneous improvement over time, rendering between patient comparisons particularly susceptible to confounding. Designs in which participants serve as their own controls reduce inter individual variability and permit more accurate discrimination between treatment effects and the natural trajectory of scar maturation. Such strategies have been formally recommended for scar prevention and reduction research to ensure that treated and control wounds would otherwise have followed comparable healing pathways [51], and have been effectively applied in clinical trials, including studies of incisional negative pressure wound therapy [52]. Whenever feasible, anatomically matched or split body models should be incorporated to enhance internal validity and more precisely define the therapeutic contribution of PRP.

For Clinical Practice, the current evidence is insufficient to support the routine use of PRP for modulation of immature post-epithelialization scars to prevent pathological outcomes. While PRP has a well-established safety profile and robust evidence for promoting healing in open wounds prior to epithelialization, data specific to biologically active but closed scars remain extremely limited. Nevertheless, the preliminary findings from the mapped studies, combined with strong biological plausibility, justify its consideration within structured clinical registries or pilot studies. Clinicians should recognize the distinction between interventions in immature post-epithelialization scars and those applied to open wounds. Within a shared decision-making framework that clearly communicates the experimental nature of the intervention and the scarcity of evidence in immature scars, PRP may be considered an off-label option for motivated patients, provided that administration is accompanied by careful documentation and objective outcome monitoring.

## Limitations of this scoping review

This review has several limitations inherent to its scope and methodology. Only five electronic databases were searched, and although no language restrictions were applied, search terms were applied in English, Spanish, and Portuguese, which may have limited identification of relevant studies published in other languages or regions. Gray literature was not included, potentially omitting unpublished or non-peer-reviewed evidence. Pediatric populations were excluded due to distinct wound-healing physiology, limiting generalizability. Finally, the review specifically focused on post-traumatic scars to deliberately exclude acne scars and striae distensae, which have distinct pathophysiology and remodeling processes [20,21].

## Conclusion

This scoping review systematically maps the highly limited and heterogeneous evidence for platelet-rich plasma (PRP) in immature post-traumatic scars and early keloids. The primary finding is a severe scarcity of dedicated research, with only five studies meeting inclusion criteria from over 1,300 screened records. Among these, two RCTs evaluated PRP for treating early keloids, while the remaining observational reports described its adjunctive use in other immature scar types, such as post-burn and facial scars. This collective evidence, while suggesting possible biological effects and a favorable safety profile, remains exploratory and insufficient to define therapeutic efficacy or guide preventive clinical practice.

Consequently, this review empirically identifies a critical translational gap between the biological rationale for early scar modulation and the current clinical research landscape. Future research must prioritize well-designed, prospective trials that directly investigate PRP's hypothesized potential to influence scar remodeling and prevent pathological scarring, which remains unproven in clinical studies, when applied during the dynamic, early post-epithelialization phase.

## Supporting information

**S1 Appendix. Complete screening process.**
(XLSX)

**S1 Text. PRISMA-ScR checklist.**
(DOCX)

**S2 Text. Completed risk of bias assessment questionnaires.**
(DOCX)

**S1 Table. Search strategy overview and results across electronic databases.** This table provides detailed electronic search strategies, databases, search dates, and number of records retrieved for studies evaluating platelet-rich plasma (PRP) and related platelet-derived products for early post-traumatic scar formation and wound healing. The search combined controlled vocabulary and free-text terms for PRP, scar types, tissue repair, and early phases of healing. Additional records were identified through backward and forward citation tracking. The total number of records retrieved across all sources was 1,325.
(DOCX)

**S2 Table. Summary of article inclusion, deduplication, and exclusion during the selection process.** This table summarizes the full selection process for studies included in the review. It reports the total number of records retrieved before deduplication, the number of duplicate and unique records identified, and the distribution of exclusions at the title and abstract screening stage. Exclusion reasons are provided according to predefined categories, including population outside the predefined timeframe (>6 months), review/editorial articles, conference abstracts, non–peer-reviewed sources, animal studies, absence of platelet-rich plasma (PRP) intervention, non-autologous PRP, insufficient data, pediatric-only populations, lack of full-text availability, and inappropriate study design. Articles meeting eligibility criteria at this stage were included for full-text review.
(DOCX)

**S3 Table. Full-text review inclusion and exclusion.** This table summarizes the outcomes of the full-text screening phase. It reports the number of articles excluded after full-text assessment along with the reasons for exclusion, based on predefined eligibility criteria, including population outside the predefined timeframe (>6 months), study design, intervention characteristics, and availability of full text. Articles meeting all inclusion criteria were included for final qualitative synthesis. The table provides a quantitative overview of inclusion and exclusion decisions at the full-text screening stage.
(DOCX)

**S4 Table. Summary of backward and forward citation searches across databases.** This table reports the results of backward and forward citation searches performed on selected source articles to identify additional studies potentially eligible for inclusion. For each round, the table provides the source article ID, date of the search (forward searches only), search type (backward or forward), total results retrieved, number of new articles selected for detailed review, number of articles meeting inclusion criteria, and reasons for exclusion when applicable. N/A indicates that no comment was provided or that the information was not applicable. Exclusion reasons include wrong population (>6 months), duplicate, review/editorial, no PRP intervention, non-autologous intervention, no full text, or other study-specific criteria.
(DOCX)

**S5 Table. Data extraction dictionary for charting included studies.** This table provides a comprehensive description of the variables used for data extraction in this scoping review. For each variable, definitions, instructions for data entry, and examples are provided to ensure standardized, reproducible, and transparent charting across studies. Variables cover study identification, design, population, PRP intervention specifics, scar characteristics, treatment regimen, outcome

measures, follow-up duration, key findings, quantitative efficacy data, adjunct therapies, methodological limitations, adverse events, and additional notes. This dictionary was developed to guide reviewers in systematically extracting relevant data for qualitative synthesis.
(DOCX)

**S6 Table. Chronological summary of included studies with PRP interventions for early and immature scars.** This table summarizes the main characteristics, interventions, outcomes, and limitations of studies included in the scoping review, presented in chronological order. Columns provide study reference, country, study design, population characteristics, scar type and timing, PRP type and preparation, route and dosing, outcome measures, follow-up duration, key findings including quantitative efficacy data when available, combination or adjunct therapies, and study limitations including adverse events. Abbreviations used in the table are as follows: PRP, platelet-rich plasma; L-PRP, Leukocyte- and platelet-rich plasma; ACP, autologous conditioned plasma; VSS, Vancouver Scar Scale; POSAS, Patient and Observer Scar Assessment Scale; BTX-A, botulinum toxin type A; FU, follow-up; CaCl$_2$, calcium chloride; Ca gluconate, calcium gluconate; RCT, randomized controlled trial; F, female; M, male; wks, weeks; mo, months; y, years; intralesional, injection into scar tissue; and topical, applied on scar surface.
(DOCX)

**S7 Table. JBI critical appraisal checklist for case reports – Ruiz et al., 2024.** This table summarizes the critical appraisal of the case report (Aura Ruiz et al., 2024) using the Joanna Briggs Institute (JBI) checklist for case reports. For each domain, "Yes" indicates the criterion was fully addressed, "No" indicates it was not, "Unclear" indicates insufficient information, and "Not applicable" applies when the criterion does not pertain to the study. The checklist evaluates patient demographics, history, clinical presentation, diagnostic assessments, interventions, post-intervention outcomes, adverse events, and clinical lessons.
(DOCX)

**S8 Table. JBI critical appraisal of case series – Kim et al., 2024.** This checklist assesses methodological quality across ten key domains, including inclusion criteria, measurement and identification of the condition, participant demographics, reporting of clinical information, follow-up outcomes, site characteristics, and appropriateness of statistical analysis. Responses are coded as Yes, No, Unclear, or Not applicable. This assessment is intended to inform evidence mapping and interpretation of study findings in the context of a scoping review.
(DOCX)

**S9 Table. Cochrane RoB 2 assessment – Albalat et al., 2022.** The table presents the five domains of bias assessed according to the RoB 2 tool: Domain 1 – randomization process, Domain 2 – deviations from intended interventions, Domain 3 – missing outcome data, Domain 4 – measurement of the outcome, and Domain 5 – selection of the reported result, along with the overall risk of bias judgment. Each domain includes the key signalling questions used to guide assessment and the final judgment (Some concerns). The predicted direction of bias indicates the potential influence of methodological limitations on study outcomes, where "Favors experimental" suggests that limitations may have overestimated the effect of the experimental intervention, and "Unpredictable" indicates uncertainty regarding the direction of bias.
(DOCX)

**S10 Table. Cochrane RoB 2 assessment – Neinaa et al., 2021.** The table presents the five domains of bias assessed according to the RoB 2 tool: Domain 1 – randomization process, Domain 2 – deviations from intended interventions, Domain 3 – missing outcome data, Domain 4 – measurement of the outcome, and Domain 5 – selection of the reported result, along with the overall risk of bias judgment. Each domain includes the key signalling questions used to guide assessment and the final judgment (Low risk, Some concerns). The predicted direction of bias indicates the potential influence of methodological limitations on study outcomes, where "Unpredictable" suggests uncertainty regarding the direction

of bias. Reviewer comments summarize the rationale for judgments, including randomization, allocation concealment, blinding of participants and assessors, completeness of outcome data, and pre-specification of statistical analyses. Overall, the study shows robust randomization and outcome measurement, but residual concerns arise from lack of participant blinding and per-protocol analysis, resulting in an overall risk of bias rated as Some concerns.
(DOCX)

**S11 Table. ROBINS-I risk of bias assessment – El-Orabi et al., 2022.** The table presents the seven domains of bias assessed according to the ROBINS-I tool for non-randomized studies: Bias due to confounding, Bias in classification of interventions, Bias in selection of participants, Bias due to missing data, Bias in measurement of outcomes, Bias in selection of the reported result, and the Overall risk of bias. Each domain includes the judgement (Low, Moderate, Serious, or Critical), the key rationale supporting the assessment, and the predicted direction of bias. The predicted direction of bias indicates the potential influence of methodological limitations on study outcomes: "Towards null" suggests an underestimation of treatment effect, "Unpredictable" indicates uncertainty regarding the direction of bias, and "NA" indicates the domain is not applicable. This table provides a transparent overview of methodological strengths and limitations in assessing the comparative efficacy of non-surgical interventions for immature facial scars.
(DOCX)

## Acknowledgments

The authors have no acknowledgments to declare.

## Author contributions

**Conceptualization:** Virgilio Blandon, Alessandro Alvarado, Miguel Borge, Erick Correa, Taylys Leyton, Sofia Gonzalez.

**Data curation:** Virgilio Blandon, Alessandro Alvarado, Miguel Borge, Erick Correa, Taylys Leyton.

**Formal analysis:** Virgilio Blandon, Alessandro Alvarado, Miguel Borge, Erick Correa, Taylys Leyton, Sofia Gonzalez.

**Investigation:** Virgilio Blandon, Alessandro Alvarado, Miguel Borge, Erick Correa, Taylys Leyton, Sofia Gonzalez.

**Methodology:** Virgilio Blandon, Alessandro Alvarado, Miguel Borge, Erick Correa, Taylys Leyton, Sofia Gonzalez.

**Project administration:** Virgilio Blandon.

**Resources:** Virgilio Blandon.

**Software:** Virgilio Blandon, Alessandro Alvarado, Miguel Borge, Erick Correa, Taylys Leyton.

**Supervision:** Virgilio Blandon.

**Validation:** Virgilio Blandon, Alessandro Alvarado, Miguel Borge, Erick Correa, Taylys Leyton, Sofia Gonzalez.

**Visualization:** Virgilio Blandon, Alessandro Alvarado, Miguel Borge, Erick Correa, Taylys Leyton, Sofia Gonzalez.

**Writing – original draft:** Virgilio Blandon, Alessandro Alvarado, Miguel Borge, Erick Correa, Taylys Leyton, Sofia Gonzalez.

**Writing – review & editing:** Virgilio Blandon.

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
