## [Decision Letter · Decision Letter 0]

10 Feb 2026

Dear Dr. Blandon,

Thank you for submitting your manuscript to PLOS ONE. After careful consideration, we feel that it has merit but does not fully meet PLOS ONE’s publication criteria as it currently stands. Therefore, we invite you to submit a revised version of the manuscript that addresses the points raised during the review process.

We look forward to receiving your revised manuscript.

Kind regards,

Anju George

Academic Editor

PLOS One

**Journal Requirements:**

**Additional Editor Comments:**

This is a methodologically sound, conceptually sophisticated, and intellectually honest scoping review regarding the limitations of the available evidence. The study not only maps existing literature but also clearly delineates an emerging field of investigation, offering valuable guidance for future research.

Reviewer 1 comments:

Scar characterization

Would it be possible to include a summary table describing how included studies defined and characterized scars (time since injury, clinical morphology, assessment scales), to make the heterogeneity of the therapeutic target more explicit?

Therapeutic window

Did the authors identify any consistent relationship between timing of PRP application (post-epithelialization interval) and reported outcomes, or could a biologically plausible therapeutic window be further discussed based on the presented pathophysiology?

PRP standardization

Is there sufficient information in the included studies to allow at least a minimal functional categorization of PRP (e.g., leukocyte-rich vs. leukocyte-poor; activated vs. non-activated), and to discuss how this variability may have influenced outcomes?

Combined interventions

How do the authors interpret the impact of frequent PRP combination with other therapeutic modalities on attribution of observed clinical effects?

Future research direction

Based on the mapping performed, which methodological elements (scar type, PRP characterization, timing of intervention, outcome measures) should be prioritized and standardized in future studies to advance clinical evidence?

Reviewers' comments:

Reviewer's Responses to Questions

**Comments to the Author**

1. Is the manuscript technically sound, and do the data support the conclusions?

Reviewer #1: Yes

2. Has the statistical analysis been performed appropriately and rigorously?

Reviewer #1: Yes

3. Have the authors made all data underlying the findings in their manuscript fully available?

Reviewer #1: Yes

4. Is the manuscript presented in an intelligible fashion and written in standard English?

Reviewer #1: Yes

Reviewer #1: 1. Title – objective – conclusion relationship

The title is precise, well delimited, and accurately reflects the scope of the study. The objective is clearly formulated as mapping the available clinical evidence on the use of PRP in immature post-traumatic scars and early keloids, as well as identifying research gaps. The conclusion is coherent with the objective and appropriately cautious, explicitly acknowledging the scarcity and heterogeneity of the evidence, without inappropriate extrapolation to routine clinical recommendations.

2. Abstract

The abstract is well structured and informative, in accordance with PLOS ONE standards. It presents a clear biological rationale, a well-defined objective, a methodology appropriate for a scoping review, summarized results with basic quantitative data (number of studies and patients), and a conclusion aligned with the level of evidence. The introductory rationale could be slightly more concise, but overall the abstract fulfills its informative purpose adequately.

3. Introduction

The introduction is one of the strengths of the manuscript. It provides a deep and up-to-date conceptual review of wound healing biology, clearly distinguishing immature from mature scars, and offers solid support for the concept of an early therapeutic “window.” The pathophysiological differentiation between post-traumatic, acne-related, and stretch mark scars is well developed and justifies the methodological choices of the study. The rationale for PRP use is biologically plausible and appropriately presented as a hypothesis not yet clinically confirmed.

4. Methods

The study design is correctly defined as a scoping review, following the frameworks of Arksey & O’Malley and Levac et al., with explicit adherence to PRISMA-ScR. The protocol was prospectively registered in the OSF, which enhances transparency and methodological rigor.

Inclusion and exclusion criteria are clear, biologically justified, and consistent with the study objective, particularly the operational definition of immature scars (≤6 months post-epithelialization). The search strategy is comprehensive, involving multiple databases and citation tracking. The processes of independent reviewer selection, resolution of disagreements, and standardized data extraction are appropriately described.

Risk of bias assessment, although optional in scoping reviews, was conducted in an appropriate and well-contextualized manner, further strengthening the methodological quality of the work.

5. Results

The results are presented in a clear, logical, and transparent manner. The PRISMA-ScR flow diagram effectively illustrates the extreme scarcity of eligible studies (5 studies out of more than 1,300 records). The characteristics of the included studies are well organized in tables, allowing rapid understanding of heterogeneity in study design, populations, interventions, PRP protocols, and outcomes.

The manuscript is appropriately cautious in avoiding quantitative or inferential synthesis, explicitly acknowledging the impossibility of direct comparisons or efficacy conclusions due to heterogeneity and the frequent use of combined therapies.

The inadequate and heterogeneous characterization of scars in the included studies limits the depth and strength of conclusions but does not compromise the methodological validity of the authors’ analysis, because:

the study is a scoping review (not a systematic review or meta-analysis);

its primary objective is to map existing evidence and identify gaps, rather than demonstrate efficacy.

Across the included studies, there is also a clear lack of standardization regarding PRP use. Substantial heterogeneity was observed in PRP acquisition and preparation methods, including centrifugation protocols, platelet concentration, leukocyte content, activation strategies, and product characterization, which were often insufficiently reported. Application protocols varied widely in terms of number of sessions, injected volume, treatment intervals, and frequent combination with other therapies. Post-treatment assessment was predominantly subjective and non-standardized, with heterogeneous follow-up and limited use of objective outcomes or validated instruments. This methodological variability limits comparability across studies and reinforces the need for standardized PRP and outcome assessment protocols in future research.

6. Discussion

The discussion is extensive, well structured, and highly qualified from a conceptual standpoint. The manuscript successfully integrates wound healing biology, scar mechanobiology, and the limited available clinical findings in a mature and balanced manner. A particularly positive aspect is the clear identification of a translational gap between biological plausibility and clinical evidence.

The authors explicitly acknowledge limitations of the existing literature, including small sample sizes, methodological heterogeneity, lack of PRP standardization, and predominantly subjective outcomes. The discussion goes beyond descriptive reporting by proposing clear directions for future research, with methodological suggestions aligned with the biology of immature scars.

7. Other observations

The writing is consistent, scientific, and appropriate for PLOS ONE. References are current, relevant, and well integrated into the text. Supplementary tables are extensive, well constructed, and add substantial value to transparency and reproducibility. No relevant conceptual inconsistencies were identified.

8. Final consideration of the reviewer

This is a methodologically sound, conceptually sophisticated, and intellectually honest scoping review regarding the limitations of the available evidence. The study not only maps existing literature but also clearly delineates an emerging field of investigation, offering valuable guidance for future research.

Suggestions to further strengthen the manuscript:

Scar characterization

Would it be possible to include a summary table describing how included studies defined and characterized scars (time since injury, clinical morphology, assessment scales), to make the heterogeneity of the therapeutic target more explicit?

Therapeutic window

Did the authors identify any consistent relationship between timing of PRP application (post-epithelialization interval) and reported outcomes, or could a biologically plausible therapeutic window be further discussed based on the presented pathophysiology?

PRP standardization

Is there sufficient information in the included studies to allow at least a minimal functional categorization of PRP (e.g., leukocyte-rich vs. leukocyte-poor; activated vs. non-activated), and to discuss how this variability may have influenced outcomes?

Combined interventions

How do the authors interpret the impact of frequent PRP combination with other therapeutic modalities on attribution of observed clinical effects?

Future research direction

Based on the mapping performed, which methodological elements (scar type, PRP characterization, timing of intervention, outcome measures) should be prioritized and standardized in future studies to advance clinical evidence?

.

Reviewer #1: **Yes:** Felipe Contoli IsoldiFelipe Contoli IsoldiFelipe Contoli IsoldiFelipe Contoli Isoldi

---

## [Author Response · Author response to Decision Letter 1]

19 Feb 2026

Dear Editor,

Thank you for the opportunity to revise and resubmit our manuscript entitled " Platelet-rich plasma for immature post-traumatic scars and early keloids: A scoping review" (Manuscript ID: PONE-D-26-02151) for consideration in PLOS ONE.

We are sincerely grateful to you and the reviewers for the insightful and constructive comments. The feedback has been invaluable in strengthening the conceptual depth and methodological clarity of our work. We have carefully addressed each point raised and have provided a detailed, point-by-point response below.

All changes have been marked in yellow in the revised manuscript file, titled 'Revised Manuscript with Track Changes' for easy identification. We believe the revisions have substantially improved the manuscript and hope it now meets the high standards of PLOS ONE.

Sincerely,

Virgilio Blandon, M.D

Universidad Americana (UAM)

virgilio.blandon@uamv.edu.ni

Response to Academic Editor

Comment: This is a methodologically sound, conceptually sophisticated, and intellectually honest scoping review regarding the limitations of the available evidence. The study not only maps existing literature but also clearly delineates an emerging field of investigation, offering valuable guidance for future research.

Response: We thank the Academic Editor for this positive and encouraging assessment of our work. We have carefully considered the subsequent comments from Reviewer 1 and have revised the manuscript accordingly to further strengthen its clarity and utility for future research.

We have also updated the PRISMA checklist to reflect the current page numbers and section locations following the revisions.

Response to Reviewer 1

We are deeply grateful for your thorough and thoughtful review. Your detailed suggestions have helped us make the manuscript more explicit and useful for the field. Below, we address each of your comments point by point.

Comment 1 (Scar characterization):

Would it be possible to include a summary table describing how included studies defined and characterized scars (time since injury, clinical morphology, assessment scales), to make the heterogeneity of the therapeutic target more explicit?

Response 1:

Thank you for this excellent suggestion. We agree that making the heterogeneity of the therapeutic target explicit is crucial for understanding the limitations of the current evidence. In response, we have created a comprehensive new table (Table 2) titled "Scar Characteristics and Assessment Methods in PRP Studies." This table provides a study-level synthesis of scar type, time since injury or intervention, morphology/clinical features, and all assessment methods used (including validated scales, dermoscopy, histopathology, and photographic documentation). This table is now presented in the Results section and highlights the variability in how scars were defined and evaluated, underscoring the need for standardized criteria in future research. Additionally, we have now incorporated a detailed analysis of scar chronicity definitions into the Results section to further emphasize this critical source of heterogeneity and its implications for interpreting treatment outcomes (lines 337-352).

Comment 2 (Therapeutic window):

Did the authors identify any consistent relationship between timing of PRP application (post-epithelialization interval) and reported outcomes, or could a biologically plausible therapeutic window be further discussed based on the presented pathophysiology?

Response 2:

This is a critical question. As noted in our review, the available data do not permit the identification of a consistent relationship between timing and outcomes due to the heterogeneity in reporting and the lack of studies designed to test this variable directly.

However, following your suggestion, we have significantly expanded the Discussion to elaborate on a biologically plausible therapeutic window based on the underlying pathophysiology. In the subsection titled " Principal interpretation: Confirmation of a paradigm gap”, we now integrate experimental evidence on PRP mechanisms (modulation of inflammation, profibrotic signaling, and neocollagenesis) with the natural history of scar maturation. We propose that the period of greatest therapeutic opportunity likely lies within the first three months post-epithelialization, when fibrotic remodeling is still evolving. We also explicitly state that this construct remains hypothetical and requires empirical validation through studies designed to vary the timing of PRP administration. This addition can be found in the revised Discussion (lines 460-482).

Furthermore, we have deepened this analysis by arguing that the evidence base is conceptually misaligned with the question of therapeutic timing. Although PRP was applied within six months post-epithelialization, inconsistent definitions of scar chronicity reflect a fundamental problem: the field lacks a shared framework for defining an "immature scar" as a preventive target. Because scar maturation stage was neither uniformly defined nor measured, dose-timing relationships cannot be analyzed. We conclude this gap is not merely empirical but ontological, as the category of "immature scar" has not been stabilized as a researchable object across studies (lines 448-459).

Comment 3 (PRP standardization):

Is there sufficient information in the included studies to allow at least a minimal functional categorization of PRP (e.g., leukocyte-rich vs. leukocyte-poor; activated vs. non-activated), and to discuss how this variability may have influenced outcomes?

Response 3:

Thank you for raising this important point. We have now addressed this directly in the Results section. We conducted a detailed re-evaluation of the included studies and found that only one study (Neinaa et al., 2021) provided sufficient detail to allow for a minimal functional categorization, specifying the use of leukocyte-rich PRP with calcium gluconate activation. The remaining four studies did not specify leukocyte content or platelet concentration, and details regarding activation were omitted or unclear. We have added this critical finding to the Results (lines 309-315) to emphasize that a consistent categorization was not feasible.

Furthermore, in the Discussion, we have expanded our analysis of technical variability. We now reference recent proteomic studies (new references [54–56]) to illustrate how subtle differences in PRP composition (e.g., leukocyte content, activation method) can lead to distinct cellular responses, ranging from anti-inflammatory M2 polarization to pro-fibrotic signaling. This discussion reinforces why the lack of standardization in the included studies fundamentally limits cross-study comparison and mechanistic interpretation (lines 508-542).

Comment 4 (Combined interventions):

How do the authors interpret the impact of frequent PRP combination with other therapeutic modalities on attribution of observed clinical effects?

Response 4:

We agree that this is a major limitation of the current evidence base. We have now made this interpretation more explicit and prominent in the Discussion. In the paragraph addressing methodological heterogeneity, we clearly state that only two randomized trials evaluated PRP as monotherapy, while the remaining studies incorporated it into combination regimens (laser, injectables, botulinum toxin). We emphasize that this approach, while reflecting clinical reality, substantially limits causal inference, as observed improvements cannot be confidently attributed to PRP alone. They may reflect the concomitant intervention, synergistic effects, or natural scar maturation. We have added a specific statement that, in the absence of controlled comparisons isolating PRP, its specific therapeutic value remains uncertain. This clarification can be found in the revised Discussion (lines [543-554).

Comment 5 (Future research direction):

Based on the mapping performed, which methodological elements (scar type, PRP characterization, timing of intervention, outcome measures) should be prioritized and standardized in future studies to advance clinical evidence?

Response 5:

Thank you for this forward-looking question. To address it comprehensively, we reorganized the dedicated subsection at the end of the Discussion titled: "Methodological Priorities for Future Research." In this section, we outline four specific priorities that emerge directly from the gaps identified in our review for future researchs (lines 565-616):

1. Rigorous Scar Characterization: Adoption of a consensus-based, operationalized definition of scar immaturity (≤6 months post-epithelialization), with explicit reporting of the interval between injury, epithelialization, and treatment initiation.

2. Transparent PRP Reporting: Mandatory reporting of absolute platelet count, leukocyte content, activation method, and centrifugation parameters. We now explicitly recommend the use of structured frameworks such as the PAW classification system [58] to enhance reproducibility.

3. Multidimensional Outcome Assessment: Integration of validated clinical scales (POSAS, VSS) with objective quantitative methodologies, including high-frequency ultrasound, 3D imaging, and elastography, to capture changes in dermal architecture and biomechanical properties.

4. Intra-individual Study Designs: A preference for split-body or intra-patient randomized controlled designs to control for confounding by patient-specific factors and the natural trajectory of scar maturation, as recommended in the literature [51,52].

We believe these priorities provide a clear and actionable roadmap for advancing the field.

We hope that our revisions thoroughly address all the points raised. We are confident that the manuscript is now stronger and more useful for guiding future research in this promising area.

Thank you again for your valuable time and expertise.

Sincerely,

The Authors

---

## [Editor Report · Decision Letter 1]

10 Mar 2026

Platelet-rich plasma for immature post-traumatic scars and early keloids: A scoping review

PONE-D-26-02151R1

Dear Dr.Blandon,

We’re pleased to inform you that your manuscript has been judged scientifically suitable for publication and will be formally accepted for publication once it meets all outstanding technical requirements.

Kind regards,

Anju George

Academic Editor

PLOS One
---

## [Editor Report · Acceptance letter]

PONE-D-26-02151R1

PLOS One

Dear Dr. Blandon,

I'm pleased to inform you that your manuscript has been deemed suitable for publication in PLOS One. Congratulations! Your manuscript is now being handed over to our production team.

Kind regards,

on behalf of

Dr. Anju George

Academic Editor

PLOS One